# Contribution of Müller Cells in the Diabetic Retinopathy Development: Focus on Oxidative Stress and Inflammation

**DOI:** 10.3390/antiox11040617

**Published:** 2022-03-23

**Authors:** Raul Carpi-Santos, Ricardo A. de Melo Reis, Flávia Carvalho Alcantara Gomes, Karin C. Calaza

**Affiliations:** 1Instituto de Ciências Biomédicas, Universidade Federal do Rio de Janeiro, Rio de Janeiro 21941-902, RJ, Brazil; rsantos@br.aspenpharma.com (R.C.-S.); fgomes@icb.ufrj.br (F.C.A.G.); 2Instituto de Biofísica Carlos Chagas Filho, Universidade Federal do Rio de Janeiro, Rio de Janeiro 21941-902, RJ, Brazil; ramreis@biof.ufrj.br; 3Instituto de Biologia, Departamento de Neurobiologia, Universidade Federal Fluminense, Niteroi 24210-201, RJ, Brazil

**Keywords:** Müller glia, Nrf2, diabetes, retina, antioxidants, reactive oxidative stress

## Abstract

Diabetic retinopathy is a neurovascular complication of diabetes and the main cause of vision loss in adults. Glial cells have a key role in maintenance of central nervous system homeostasis. In the retina, the predominant element is the Müller cell, a specialized cell with radial morphology that spans all retinal layers and influences the function of the entire retinal circuitry. Müller cells provide metabolic support, regulation of extracellular composition, synaptic activity control, structural organization of the blood–retina barrier, antioxidant activity, and trophic support, among other roles. Therefore, impairments of Müller actions lead to retinal malfunctions. Accordingly, increasing evidence indicates that Müller cells are affected in diabetic retinopathy and may contribute to the severity of the disease. Here, we will survey recently described alterations in Müller cell functions and cellular events that contribute to diabetic retinopathy, especially related to oxidative stress and inflammation. This review sheds light on Müller cells as potential therapeutic targets of this disease.

## 1. Pathophysiology of Diabetic Retinopathy

Diabetic retinopathy (DR) is a neurovascular disease that ultimately leads to blindness in diabetes mellitus patients [1]. Visual loss impacts the quality of life, daily activities, and reduces individual work capacity [2], contributing to economic burden on health systems [3]. Due to the absence of a cure and the unavailability of efficient treatment for DR, development of therapeutic alternatives should be a priority. Emerging evidence points to the role of Müller glia cells (MGC), not only as crucial players in the retinal physiology, but also involved in many dysfunctional aspects in diabetes. This review aims to discuss the molecular and cellular mechanisms underlying DR, focusing on early retinal alterations in the diabetic condition as a promising therapeutic window to prevent blindness. Here, we will present evidence of the contribution of MGC deficit to DR progression, and we will suggest MGC as alternative therapeutic target to DR.

Glycaemia uncontrolled over a long period of time, dysregulates several metabolic pathways in the retina, leading to a progressive neurovascular impairment characterized by many elements, such as alteration in the levels of growth and neurotrophic factors, oxidative stress, inflammation, hypoxia, adhesion molecules, and genetic factors [4,5,6,7]. Since diagnosis and treatment of DR rely on the detection of microvascular alterations, it is considered a vascular disease, divided into two late stages: non-proliferative diabetic retinopathy and proliferative diabetic retinopathy. A large amount of data gathered over the last two decades has revealed many early modifications in neural aspects of retinal physiology/anatomy, extending the current concept of DR so as to be considered a neurovascular disease. Whether diabetic retinal neuropathy is the cause or effect of microangiopathy under chronic hyperglycemia is still under debate [8,9].

The retina is a special tissue with a highly ordered architecture containing light transducing cells (photoreceptors), interneurons, projection neurons, and three types of glial cells. The cell bodies of retinal cells are organized in three nuclear layers intercalated by plexiform layers, consisting of cell processes and synapses (Figure 1A). Located on the outermost layer, closer to the choroid, are two types of photoreceptor cells, approximately 120 million rods [10] and 4.6 million cones in the human eye [11], responsible for phototransduction, and forming the outer nuclear layer. In addition, around 0.7–1.5 million retinal ganglion cells, found in the ganglion cell layer, transmit electrical activity through the optic nerve to the brain’s visual processing areas [12]. In between these two layers, there is the inner nuclear layer, containing the cell bodies of excitatory bipolar interneurons, controlled by inhibitory horizontal and amacrine neurons, and MGC, a fundamental cell involved with major functions in the retina.

The retina is known to be one of the most energetically demanding tissues in the body [13]. The energy support for the mammalian retina comes from the inner and outer vascular systems. The vascularization of the inner retina consists of three parallel planar vascular plexuses that originate from a branch of the internal carotid artery, the ophthalmic artery. The ophthalmic artery enters through the optical disc and branches into the inner (ganglion cell layer) and outer (inner plexiform layer, inner nuclear layer to outer plexiform layer) intraretinal bed and the peripapillary bed, present in the human and mouse eye but absent in the rat [14]. The blood supply for the outer retina comes from the choroidal vasculature. The intraretinal and the choroid blood vessels consist of, respectively, non-fenestrated and fenestrated endothelial cells, the former surrounded by pericytes. Therefore, the inner, but not the outer, retinal vasculature has a blood–retinal barrier (BRB) that rigorously regulates the molecular traffic between blood and retina.

A hallmark of the initial non-proliferative diabetic retinopathy vascular damage is the induction of basal membrane thickness. This leads to a reduced endothelial adherence and tight junction impairment, resulting in BRB breakdown. Moreover, there is cell loss of pericytes and endothelial dysfunction, resulting in fragile capillaries, formation of microaneurysms, vascular leakage, and small hemorrhages. The accumulation of these vascular lesions results in ischemia in some areas of the retinal tissue. The ischemic condition induces a further increase in the release of pro-angiogenic factors, such as vascular endothelial growth factor (VEGF), promoting neovascularization, a hallmark for the beginning of proliferative diabetic retinopathy [15]. However, these new blood vessels, in most cases, are not functional, aggravating the damage. Neovascularization exacerbates micro-hemorrhages and can also lead to retinal detachment, exacerbating the patient’s visual loss [16].

Although the current diagnosis of DR is based on vascular alterations, and treatment is limited to more advanced stages of the disease, a large amount of data has revealed changes in neural retina before clinical signs of vascular abnormalities in animal models and patients [17,18,19,20]. Indeed, electrophysiological studies have demonstrated that diabetic patients present neuronal alterations even when no vascular (clinical) signs are detected in the examination of the eye fundus [21,22,23]. 

Neurodegeneration is one of the earliest signs that occurs in experimental models for DR, detected as early as two weeks post-diabetes induction [17,24,25,26]. Accordingly, Carrasco and coworkers (2007) have described that post-mortem diabetic patients with no clinically detectable signs of DR show a significantly higher number of apoptotic cells in all retinal layers [18]. More recently, with the advance of powerful methodological strategies, such as optical coherence tomography, it has been shown that patients with diabetes have thinner inner nuclear–ganglion cell layers and reduced nerve fiber layer, confirming in an in vivo procedure that neurodegeneration is an early event in DR pathology [27,28,29]. Indeed, a thinner nerve fiber layer has been correlated with early changes in the peripapillary vessel morphology and vessel density of the radial peripapillary plexus in patients with diabetes without DR. Earlier changes in superficial vessel density have been documented in the peripapillary rather than in the macular region [30]. Based on these new imaging techniques new biomarkers have been developed that could help in the detection of subclinical disease, before clinically (vascular) detectable changes or visual symptoms are noted [31]. However, this approach is expensive, which may limit the impact in preventing DR worldwide.

In addition to neurons, three types of glia can be found in the retina: astrocytes in the nerve fiber layer; microglia mainly in the plexiform layers; and, the predominant type, the MGC. The optic nerve presents oligodendrocytes besides astrocytes and microglia. We have previously shown that all these glial types are altered in the type I diabetes mice model, and attenuation of inflammation controls this response [32]. 

Glial cells have key functions in maintaining brain homeostasis and in the control of central nervous system (CNS) in response to insults. Emerging evidence from our research and others supports the concept that reactive glial cells play fundamental roles in neurodegenerative diseases such as Alzheimer’s and Parkinson’s diseases [33,34,35,36,37]. In common, these conditions present a general neuroinflammatory scenario, characterized by production and release of pro- and/or anti-inflammatory cytokines and chemokines, antioxidants, free radicals, and neurotrophic factors by glial cells, such as astrocytes, microglia, and MGC. Whether this response contributes to the development or delay of the disease is still a matter of discussion. Whereas involvement of glial cells in brain diseases is well documented, the contribution of MGC to neurodegenerative diseases such as DR, is less clear. Given the large number of functions performed by MGC, it is expected that alterations of these cells might have a major impact in DR outcome. In the next sections, we will present the roles of MGC in retinal function and homeostasis and discuss accumulating evidence of the involvement of these cells in DR.

## 2. Role of Müller Cells in Retinal Physiology

MGC is the predominant (90%) glial element of the vertebrate retina. It has been considered for a long time to be a passive element, supporting the structure and function of the architecture of the vertebrate retina. Identified in the middle of the 19th century by the anatomist Heinrich Müller as radial fibers [38,39], these cells span the entire retinal tissue: with the soma in the inner nuclear layer and their radial processes extending from the inner vitreous border to the outer retina in the inner segment of photoreceptors [40,41] (Figure 1A). Due to this unique morphology, the MGC interact with cells in all retinal layers contributing to specify neuron–glia–vascular–epithelium functional domains. 

Although a full assessment of MGC morphology and function is beyond the scope of the present review, we will provide an overview of MGC properties in the healthy retina, important to understand the impact of its dysfunction in DR. 

MGC play several important roles in the outer retina. These cells regenerate all-trans retinol to 11-cis retinal participating in the cone visual cycle, a crucial step to maintain the response to light, a process formerly attributed exclusively to pigmented epithelium cells [42,43]. Important for visual acuity, is the ability of MGC to function as optical fibers guiding the light [44,45,46,47,48]. Interestingly, the local density of MGC and cones are the same in many species in a way that each Müller cell guides the light to “its own” cone photoreceptor [45,48]. Thus, MGC bypass the light scattering by nerve fibers and synapses allowing minimal light distortion and little light loss to photoreceptors. The strands of microtubules and intermediate filaments in the radial fibers of MGC may also contribute to structural stability for the retina, especially for central fovea [45].

The outer processes of MGC are attached to each other and to the inner segments of photoreceptors by adherens and tight junctions to form the outer limiting membrane (OLM). The OLM plays a role in maintaining the structure of the retina through mechanical strength and as part of the outer retinal barrier [49,50]. 

The ability of MGC to phagocyte is important to the integrity of photoreceptors and for the rearrangement of new cone outer segments (COS) under physiological [51] and pathological conditions [52]. In addition, there are two types of MGC in the fovea: specialized MGC, which form a structure called the Müller cell cone in the foveola; and the z-shape MGC of the fovea walls and parafovea [40]. The specialized MGC of the foveola contains a high density of macular pigment, protecting photoreceptors by acting as a short-wavelength light filter compensating the absence of blood vessels in this region [45,53].

The inner retina also benefits from MGC [41,54]. The vitreous endfeet of MGC processes, together with an extracellular matrix structure, form the inner limiting membrane (ILM), which is critical in defining the retinal border and stabilizing ocular blood vessels [41,54]. The ILM region is the main location of TRPV4 (Transient Receptor Potential Cation Channel Subfamily V Member 4) in MGC, which seems to exert an important role as an osmosensor, crucial for volume (dys)regulation of the retina under both healthy and pathological conditions [55]. The TRPV4 function in MGC requires arachidonic acid and 5,6-EET [55]. 5,6-Epoxyeicosatrienoic acid (5,6-EET), among other inflammatory signals, activates TRPA1 [56], which is also expressed by MGC [57]. MGC also respond to swelling or shrinkage of neurons or extracellular space, opening potassium and chloride channels and adapting their morphology, regulating cellular volume [58]. Additionally, aquaporin 4 (AQP4), a water channel present in MGC, is also involved in retinal volume regulation [59]. Water in the inner retina is transported by MGC and can be eliminated by pigmented epithelium cells [60,61]. So, MGC and retinal pigment epithelium cells (RPE) actively drain water and osmolytes outside the retina. The potassium channels involved in water removal in MGC are also important for the regulation of ionic balance. MGC take up and redistribute extracellular potassium, arising during neuronal activity especially in the plexiform layers, in a process called potassium spatial buffering [62]. This is very important for maintenance of the neuronal microenvironmental since the increase in extracellular potassium induces neuronal depolarization and is also coupled to drainage function of MGC [50]. 

MGC also have a huge influence on the vascular function in the retina, from the establishment of BRB [63] to the maintenance and enhancement of its function [50,64]. Retinal arterioles, venules, and capillaries are tightly ensheathed by macroglia. In the retina, the vascular tone is controlled by glial cells, astrocytes, and Müller cells. Since the astrocyte cells are restricted to nerve fiber layers and GCL, Müller cells probably have a predominant action [65,66,67] (Table 1). ATP seems to be a key modulator, by acting as P2X receptors, inducing the production and release of metabolites of arachidonic acid that control vessels constriction [67]. The described coupling between neuronal activity and blood flow is called neurovascular coupling, which is also largely mediated by glial cells in the retina [67].

In addition, Müller cells primarily metabolize glucose by anaerobic glycolysis [131], and represent the main site of retinal glycogen storage, from which metabolites can be donated to other retinal cells [134]. Thus, oxygen is saved for neurons warranting more resistance to anoxia/hypoxia conditions and the lactate can be donated to neurons, to use in alternative energy pathways to produce ATP, especially for photoreceptors [132,133] (Table 1).

The special and characteristic morphology of MGC enabling them to transverse the retina, together with the ability to uptake and metabolize glutamate and gamma-aminobutyric acid (GABA), the main excitatory and inhibitory neurotransmitters, respectively, underlies another key role in the retina: control of neurotransmission [40]. Glutamate uptake by MGC is converted to glutamine, through glutamine synthetase, the so-called glutamate–glutamine cycle, regulating the driving force of neuronal glutamate uptake and the maintenance of biochemical pathways, especially in the inner retina [131,138]. In addition to the buffering of GABA and glutamate availability, MGC also regulate neuron excitability, especially ganglion cells, neuroprotection, and immune response through gliotransmitters (ATP, glutamate, D-serine, among others), cytokines, neurotrophic factors, antioxidants, and inflammatory signals [41,139,140].

Finally, MGC are considered as endogenous stem cells in the adult retina, and upon being activated by damage or growth factors, they proliferate, de-differentiate, and contribute to neural regeneration [113]. However, the regenerative potential of MGC varies greatly according to vertebrate species, behaving as radial-glial-like neural stem cells in fish and amphibians (regenerating retinal neurons in response to injury) and being extremely limited in mammals and birds [141]. Pathological conditions, such as excitotoxicity, induce MGC gliosis increasing GFAP expression, changing the identity and signaling pattern [113,142].

In the past two decades, various genes and pathways have been studied to support research on the stemness shown by MGC. Postnatal and adult mouse MGC have been genetically reprogrammed through the expression of proneural genes in the Lin28-let7-Ascl1 axis to unlock the neurogenic potential of mammalian MGC [143]. Overexpression of ASCL1 in dissociated mouse MGC in cultures or in intact retinal explants upregulated retinal progenitor-specific genes and downregulated glial genes [144]. MGC-derived progenitors differentiated into cells that exhibited neuronal morphologies, expressed retinal subtype-specific neuronal markers, and displayed neuron-like physiological response. Recent data have shown a generation of retinal neurons from pluripotent stem cells or MGC, thus implying that an in situ cellular reprogramming strategy for restoration of vision might not be so distant [145,146,147,148]. Alternatively, a small subset of miRNAs (highly conserved ~22 nucleotide (nt) RNAs that post-transcriptionally regulate gene expression) involved with development in vertebrates, has been recognized as controlling MGC reprogramming during retina regeneration, behaving as therapeutic targets for treatment of visual disorders and damage [149]. Recent data from the lab of Iqbal Ahmad suggest that in the near future, neurogenic properties of MGC might be revealed by serial exposure to different combinations of small molecules such as a cocktail [150]. For instance, Müller cells exposed to I-bet151, an inhibitor of the binding of BET family of proteins to the cell-type specific activated chromatin domains, forskolin, a diterpene adenylate cyclase activator, isoxazole 9 (ISX9), which induces neuronal differentiation, might silence glia-specific genes, and induce acquired transcriptional signature of neurons, therefore regulating neuronal potential of MGC. Neurons derived from MGC expressed neuronal proteins and showed electrophysiological features of immature neurons [150]. Thus, reprogramming of MGC might represent a possible treatment for regeneration of the retinal neural circuits, restoring visual capacity in patients in advanced stages of DR.

As discussed, MGC play key roles in the retinal functions summarized in Figure 1B [40,138,139,140,151,152]. The current concept of MGC implies that these cells are active players in a healthy or injured retina. Due to the broad actions of MGC in retina homeostasis, it is likely that dysfunctions on these cells can severely impact retinal function. Essentially, Müller cells respond primarily with a proinflammatory phenotype when activated and secretion of cytokines and components of the complement system [153] Authors described mitochondrial dysfunction, implying oxidative stress after treatment with the various cytokines, highlighting the importance of Müller cell signaling in the aggravated retina, indicating an active role in chronic retinal inflammation. A combination of high glucose levels and oxidative stress favor DR in Müller cells, which reduce glutamine synthetase levels, and augment the migration capacity of Müller glia. This suggests that these experimental conditions could induce some degree of dedifferentiation and favor the migration ability. Indeed, a correlation with several expressed genes and transcriptomic alteration related to glucose metabolism, cell migration, development, and pluripotency were found in cells of diabetic animals associated with the histone deacetylase sirtuin 6 (SIRT6), as an epigenetic modulator [154]. One of the transcription factors possibly involved in the differential expression program observed in diabetic MGs is SRY-Box transcription factor 9 (SOX9). Regarding inflammation, it is known that this multi-step process plays a crucial role in retinal regeneration when the model is zebrafish. Several mediators are involved as pattern recognition receptors activated by damage-associated molecular pattern molecules, including the chromatin-associated nuclear protein high mobility group 1, heat shock proteins, purine metabolites, such as adenosine triphosphate (ATP), and uric acid [155]. In the chick retina, nuclear factor kappa B (NF-κB), a critical regulator of inflammation, was shown to inhibit the reprogramming of Müller glia into proliferating progenitor cells [156]. In addition, inflammation-induced mammalian target of rapamycin (mTOR) signaling in the Müller glia was shown to be important for retina regeneration in the adult zebrafish [157] and the chick model [158]. Finally, the innate immune system also plays key roles in inflammation, reprogramming, and tissue regeneration. Indeed, microglia stimulate neurogenesis in avian and fish Müller cells after injury, but they do not normally regenerate; overexpressing the pro-neural achaete-scute family bHLH transcription factor 1 (Ascl1) in Müller glia is sufficient to stimulate regeneration into functional neurons in the adult mouse retina neurogenesis. A combination of Ascl1:Atoh1 (from atonal family bHLH transcription factor 1), increases efficiency at stimulating neurogenesis, even in the absence of retinal injury [159]. Microglia inhibit the Ascl1-mediated retinal regeneration, suggesting that the innate immune system limits the regenerative response to injury [159]. Reactive microglia coordinate signals as NF-κB to suppresses the reprogramming of Müller glia into proliferating progenitor cells [156]. In the next section, we will discuss the implications of the MGC function on the outcome of DR.

## 3. Müller Cells in DR 

As discussed previously, while there are many studies on neurons and vascular cells in DR, less attention has been given to MGC cells in DR research, especially in the early stages of the disease, when no microangiopathy is observed. Accumulating evidence from animal experimental models and human post-mortem samples, as well as patient evaluations, strongly suggests that MGC play an important role in DR outcome. By using two different experimental models, Shen and coworkers have shown that Müller depletion leads to BRB breakdown and vascular alterations, two hallmarks of DR pathology [160]. Analysis of post-mortem samples from diabetic patients with no signs of vascular alterations revealed higher levels of glial reactivity, suggesting that this is an early event in DR development and could precede the vascular clinical symptoms [18,161,162]. Patients with non-proliferative diabetic retinopathy have displayed increased thickness of the inner nuclear layer and outer plexiform layer, which may be a result of activation and Müller cell swelling [163], which involves a number of angiogenic molecules and inflammatory cytokines [164]. According to the idea that MGC are altered in diabetic conditions, hyperglycemia directly upregulates GFAP in MGC in culture, a classic signal of gliosis [165,166]. Indeed, high glucose treatment in MGC cultures results in a wide range of pathological responses. Oxidative stress (increase in reactive oxygen species (ROS) and decrease in antioxidant systems) and activation of inflammatory responses is an early response in MGC to hyperglycemia conditions [165,167,168,169,170,171,172]. These alterations lead to the activation of cell death pathways. Apoptosis is described due to a decrease in Bcl-2 and an increase in Bax (bcl2 associated X) and caspase-3 levels [173,174,175]. Recently, several studies have shown the activation of pyroptosis and mitophagy/autophagy pathways [169,176,177,178,179,180]. Corroborating these findings, various morphological and functional alterations have been described in MGC in diabetes experimental models. Increase in GFAP labelling in MGC, a hallmark of glia stress and reactivity, is detected in diabetes induction models [181,182,183]. Gliosis of the foveolar MGC is recognizable in OCT images by the presence of hyperreflective dots (HRD) [40]. Since inner processes of MGC seem to be important to compensate for the thin basal lamina of the fovea, providing structural stability, the increase in intermediate filaments by MGC may increase the resistance against tractional forces in DME [184]. However, the absence of these filaments in the outer processes of MGC may explain the disruption between inner layer and OLM in DME. Disruption of the foveolar Müller cell layer results in macular hole formation [45]. Further, diabetes also leads to a reduction in and displacement of the Kir4.1 channel and to an increase in aquaporin-4 in MGC, which can be associated with edematous alterations and retinal hyperexcitation [75,82]. Intercellular fluid accumulation in MGC provokes the cyst formation observed in DME, even though it is still in debate whether this is intracellular or extracellular [185]. In DR progression, regulation of neurotransmitters is also impaired. In retinas from diabetic animals, there is a decrease in excitatory amino acid transporter 1 (EAAT1) and glutamine synthetase expression/activity in MGC, which promote cell death by excitotoxicity from early stages of diabetes [85,93,182]. These data agree with the increase in glutamate in the retina of animal models of DR [17,88,89,90] and in diabetic patients [91].

According to the idea that MGC play a crucial role in the outer retina [45,51,52,53], a disruption in OLM and in the distance between OLM and RPE, known as outer retinal layer, observed in OCT, has been correlated with poorer visual acuity [50,185] and to poor responses to anti-VEGF therapy in DME [186,187]. Chronic DME can also lead to structural disorganization of retinal inner layers (DRIL) due to glial and neuronal cell death. DRIL is a good predictor of visual outcome [188]. Another correlation of MGC in DME has been related to the correlation of poor visual acuity and the destruction of Müller cell cones [189]. Altogether, these data indicate that MGC seem to be closely involved with DME.

Further evidence of the involvement of MGC in the development of DR has been found in studies with VEGF, oxidative stress, and retinal inflammation. MGC is the main source of VEGF in the retina [190,191], and the increase in VEGF triggered by diabetes is associated with upregulation of GFAP, a signal of gliosis [75]. Conditional knockout of VEGF in MGC reduced nearly 50% of the total VEGF in the retina of diabetic animal models and greatly prevented vascular dysfunctions and increased permeability [108,192]. Interestingly, the increase in permeability induced by VEGF is inhibited by the pigment epithelium-derived factor (PEDF) produced by MGC [193], indicating that MGC can control retinal vascular permeability. Some of the hallmarks of DR, such as increase in retinal VEGF and tumor necrosis factor alpha (TNF-α), and pericyte loss were decreased by the disruption of β-catenin in MGC of diabetic animals [194]. Oxidative stress upregulates VEGF in several retinal cell types, such as endothelial cells [195], human retinal pigmental epithelial (ARPE-19) cells [196,197], and MGC [198], decreasing cell viability. On the other hand, VEGF increases oxidative stress [199]. Alternatively, VEGF conditional knockout mice also present reduction in nitrosylated proteins [199], NFκB activation [108], intercellular adhesion molecule (ICAM), and TNF-ɑ expression [108,192,199], hallmarks of retinal inflammation. Indeed, VEGF gene polymorphisms have been associated with the risk of DR [200]. Multiple studies have described various pathways that induce an increase in VEGF in MGC [5,198] and recently it was shown that VEGF upregulation is determined by an autocrine loop. Oxidative stress induces VEGF release by MGC, which activates the VEGF receptor 2 in MGC, further strengthening both VEGF expression and oxidative stress [198]. Therefore, MGC VEGF appears to be involved in important pathological features of early stages of DR. Interestingly, VEGF is also important to MGC and all retinal neuronal types after a long period of diabetes. Conditioned knockout of VEGF receptor 2 in MGC leads to a significant decrease in BDNF and GDNF and the loss of MGC and all types of retinal neuronal cells in diabetic mice [128]. Finally, several studies have also shown the crucial correlation of oxidative stress, VEGF and vascular alterations and neovascularization, features of later stages of DR [201,202,203].

Thus, MGC-derived VEGF induces retinal inflammation, oxidative stress, vascular leakage, and neovascularization [108,192]. The temporal course of in vitro changes detected in MGC or in ganglion cells by hyperglycemia corroborates the idea that there is a connection between neural damage and oxidative stress/inflammation response in Müller glia. As early as a few hours in a hyperglycemic medium, MGC show signals of oxidative stress and inflammation [167,171,178,204]. However, the oxidative stress, inflammation, and cell death of ganglion cells appear to occur later on [205,206,207].

Interestingly, recent data have suggested other modulators, such as fibroblast growth factor type 2 (FGF2), important to Müller cell activation, which expand the range of possible therapeutic targets to be investigated [208]. Next, we will present molecular pathways by which MGC control two other essential events for DR progression: oxidative stress and inflammation.

### 3.1. Oxidative Stress

Today, it is well-recognized that oxidative stress is one of the main causes of cellular imbalance in DR [209]. The hyperglycemic state induces multiple alterations in the biochemical pathways, accumulation of advanced glycation end products, and higher citric acid cycle activity that result in increased reactive oxygen species (ROS) production in the retina [210,211,212]. 

In animal models, many supplemented natural antioxidants, such as cocoa [213], quercetin [214] -based flavonoids [215], green tea [216], and apocynin [217], have been shown be beneficial, as they inhibit apoptosis, decrease proinflammatory cytokines, and suppress endoplasmic reticulum stress and ROS production in the retina [218,219]. Although the use of a specific antioxidant molecule or a combination of antioxidants might represent a promising therapeutic strategy for DR, the number of clinical trials with general antioxidant molecules for DR treatment are scarce and the results are controversial and disappointing [220,221,222]. For example, in clinical studies investigating combined antioxidant supplementation, it was demonstrated that an eighteen-month antioxidant therapy reduced malondialdehyde (MDA) levels and increased antioxidant capacity in the retina of type 2 diabetic patients with DR [223]. Further, antioxidant therapy slowed the progress of DR stages in the supplemented group in a longer follow up of five years; the visual acuity, however, was not different between the supplemented and non-supplement groups [224]. Another example of conflicting results with clinical trials was reported with the dietary antioxidant, lutein. Lutein and its isomers are carotenoids found in a wide variety of vegetables and fruits, with particularly high concentrations in leafy green vegetables, known to be beneficial for ocular health and vision [225]. In 2017, it was demonstrated that a nine-month therapy with lutein had a beneficial effect on contrast sensitivity in non-proliferative diabetic patients [226]; however, a recent study, with more subjects, did not detect any difference in visual acuity after 4 months of supplementation [227]. Thus, even though a large number of in vitro and animal model investigations have suggested that antioxidants might prevent DR development, clinical studies have not been able to corroborate such efficiency. Whether this controversy is due to differences between humans and rodents or due to different methodological approaches remains to be investigated. A better understanding of the molecular and cellular mechanisms of antioxidants and their targets would be helpful in designing more successful therapies for DR. For example, recently, we have demonstrated that eye drop treatment with the antioxidant α-lipoic acid, an inhibitor of transient potential receptor ankyrin 1 (TRPA1) or its absence in knockout animals, prevents oxidative stress and cell death in retinal ischemic conditions [57]. This kind of molecular target could be useful for alleviating retinal oxidative stress under conditions of diabetes and the impairments associated with these circumstances.

Glial cells have been described as important controllers of oxidative stress in different brain regions. We have shown that hesperidin and casticin, flavonoid compounds, increase the neuroprotective potential of cerebral cortex glial cells in vitro and in mice by modulating the secretion of protective factors [228,229]. 

MGC play a fundamental role in retinal antioxidant defense. High glucose leads to increase in Heme Oxygenase-1 expression and ROS production in these cells [82,178,230]. Furthermore, MGC is the main cell that contains glutathione (GSH) that is released in response to tissue stress, to support other cells against oxidant insults [231,232,233]. GSH activates calcium shifts in MGC and induces GABA release that could help to decrease excitotoxicity in the retina [234]. GSH is a thiol-containing tripeptide and is one of the main intracellular antioxidant molecules, acting directly by scavenging reactive oxygen and nitrogen species [235]. We have shown that GSH levels are lowered after 3 weeks of hyperglycemia in the rat retina [236] and remain low until later stages [237,238], probably contributing to the oxidative stress status of diabetic retina.

GSH impairment is a multifactorial phenomenon; for example, a reduction in γ-glutamylcysteine synthetase, an enzyme responsible for GSH synthesis, has been reported to be reduced in DR [239]. Since GSH is a tripeptide produced with glutamate, cysteine, and glycine, the levels of intracellular cysteine are also a limiting factor for its production [240]. System x_c_^-^ is a cystine/glutamate exchanger that mediates cystine uptake. Once inside the cells, cystine is reduced to cysteine [241]. Therefore, this exchanger activity is crucial for maintaining GSH levels in the retina [242]. Our group was pioneer in demonstrating that system x_c_^-^ is affected in the retina of DR animal models [236]. Both content of the catalytic subunit (xCT) and activity of this transporter are reduced at very early stages of diabetes.

We also have shown that this transport system remains impaired up to 6 months after the diabetes has been diagnosed [237]. Since system x_c_^-^ seems to be enriched in MGC [243], these alterations could lead to a condition in which MGC could be depleted of one of the main molecules involved in control of oxidative stress under diabetic conditions. Accordingly, in the Müller cell cultures of Sigma 1 receptor in knockout mice, there is an increase in oxidative stress that occurs due to a reduction in system x_c_^-^, and decrease in GSH levels [199].

As mentioned before, treatments with antioxidant molecules have failed in clinical trials so far. Although the reasons for the failure are not completely known, one possible explanation is that the action of exogenous antioxidants might be limited by the impaired antioxidant properties of retinal cells. A possible solution for this problem comes from MGC and nuclear factor erythroid 2-related factor 2 (Nrf2), which is a key transcription factor and regulator of cellular antioxidant defense. Upon an increase in cellular ROS levels, Nrf2 is translocated to the nucleus and induces the expression of several genes associated with oxidative stress protection, including the enzymes responsible for GSH production [244,245,246], increasing cellular antioxidant capacity in response to a ROS upsurge. In particular, Nrf2 binds to a specific promoter region, an antioxidant-responsive element (ARE) that increases xCT expression, the catalytic subunit of system x_c_^-^ [247]. In the retina, Nrf2 is strongly and preferentially expressed by MGC [172]. Nrf2 has been shown to be essential for maintenance of the antioxidant status; knocking out Nrf2 increases oxidative stress directly induced by oxidants in MGC, intensifies oxidative stress and ganglion cell death, worsening retinal dysfunction in diabetic retina [172,248]. The same is seen in other models of retinal neurodegeneration [249,250]. An increase in oxidative stress in MGC is seen when Nrf2 levels and activity are reduced [251]. Hyperglycemia induces a fast decrease in nuclear Nrf2 in cultures of MGC [167,169,252]. Accordingly, hyperglycemia promotes a decrease in antioxidant defense elements (activity of catalase and superoxide dismutase (SOD), GSH, HO-1, and NAD(P)H: quinone oxireductase 1 (NQO-1) levels in MGC, intensifying oxidative stress [167,169,170,204,252] and affecting MGC viability [253]. 

Our group and others have shown that Nrf2 expression, activity, and ARE-binding oscillate with diabetes progression resulting in lower levels of system x_c_^-^ and GSH in diabetic animals [157,167,237,254]. Nrf2 was reduced after one month in diabetic rat retinas, recovered to normal levels in 2 months, and again decreased after 6 months of diabetes (Figure 2A) [237]. The expression of xCT follows this same pattern [237], and is probably a result of a decrease in xCT-induced expression by Nrf2 (Figure 2B). This suggests that the retinal tissue has a compensatory mechanism to improve the cellular antioxidant defense in response to oxidative stress, a recovery in Nrf2 activity was detected in 2 months, after an initial decrease. However, this was not sufficient to restore proper oxidative status, as seen by continuous high levels of ROS and lower GSH through the time window analyzed (Figure 2C). Upon the continuous stress inflicted by diabetes in the retina, this mechanism is overwhelmed and Nrf2 is impaired after 6 months. Thus, it is possible that we have an early time window for initial treatment with Nrf2 inductors that would be more efficient because of this compensatory response [237,239]. This is an unexpected result because if oxidative stress is a stimulus for Nrf2 activation, why is there Nrf2 impairment in DR models, which present higher oxidative stress status? A recent explanation for this phenomenon has been suggested based on the modulation of the protein, “regulated in development and DNA damage response-1” (REDD1). REDD1 is a stress induced protein, which acts as a negative regulator of Nrf2, and has its expression increased with oxidative stress in diabetic mice retina and in MGC exposed to high glucose (Miller et al., 2019, 2020). High glucose treatment induces oxidative stress in the human retinal Müller cell line (MIO-M1); however, oxidative stress is not observed in REDD1 knocked down cultures, neither is the expression of Nrf2-driven genes detected. Furthermore, it was also shown that REDD1 induces Nrf2 degradation through glycogen synthase kinase 3β (GSK) activation. Diabetic mice treated with a GSK inhibitor had higher levels of Nrf2 activity and lower oxidative stress in their retinas [255]. GSK is classically regulated by Akt, which phosphorylates and inactivates GSK [256]. However, hyperglycemia decreases Akt activity, stimulating GSK [173]. Thus, stimulated GSK increases the Nrf2 extrusion from the nucleus, inducing a decrease in nuclear Nrf2 [157]. Thus, Nrf2 impairment seems to be a crucial step to induce oxidative stress in DR while treatments that boost glial Nrf2 activity might have a benefit effect (Figure 2D). Indeed, a synthetic Nrf2 activator, dihydro-CDDO-trifluoroethyl amide (dh404), increases the antioxidant capacity of MGC exposed to high glucose, and inhibits the upregulation of angiogenic and inflammatory molecules. Further, in diabetic rats, dh404 was able to prevent the increase in oxidative stress, pro-inflammatory molecules, and BRB breakdown, major hallmarks of DR progression [168]. Other molecules that also activate MGC Nrf2 were reported to have beneficial effects in DR, such as fenofibrate [257] and sulforaphane [169]. In addition, MGC are directly affected by hyperglycemia. One of the early alterations observed in cultures of MGC was in the ROS levels [178,204], along with an increase in inflammatory signals (TNF-alpha1, iNOS, NFkB), activation of caspase-1, all observed in the first 24 h [171,178,180]. The data suggested that these changes are followed by stimulation of caspase-3 and -9, cytochrome c release, and MGC death [171,173]. MGC are directly impacted by ROS, with an increasing system x_c_^-^ activity and decreasing GLAST levels [258]. In addition, MGC also respond to oxidative stress indirectly generated by other stimuli. For instance, MGC show an increase in ROS after 1h of exposure to modified LDL, which triggers endoplasmic reticulum stress and cell death [259]. Therefore, targeting Müller oxidative stress might lead to better therapeutic treatments for prevention of oxidative stress in DR.

### 3.2. Inflammation

DR is a chronic low grade inflammatory disease [211,260], generally associated with microglia/macrophage activation in the retina with the upregulation of various proinflammatory, such as galectin-3, angiogenic factors, and cytokines, such as TNF-α and interleukin 1-beta (IL-1β) [261,262]. The levels of these particular factors, among others, have been shown to be enhanced in the aqueous humor of diabetic patients (with or without retinopathy) compared to non-diabetic controls [263]. Although conflicting results appear from several reports, inflammatory cytokines and chemokines are increased in DR in both serum as well as in the eye [264]. However, local rather than systemic production of proinflammatory cytokines seems to be a hallmark of DR. Inflammatory milieu contributes to the development of retinal damage in DR, such as BRB breakdown, neovascularization, neuronal, and capillary degeneration [163]. 

Due to the inflammatory profile of DR, treatments that inhibit the pro-inflammatory burden have been suggested as promising alternatives in DR control. In a multicenter randomized controlled clinical trial (1989), it was shown that aspirin, a nonsteroidal anti-inflammatory drug, slows the appearance of microaneurysms in diabetic patients, but had no effect in patients with advanced DR. In addition, in a study to evaluate the efficacy and safety of an IL-1β inhibitor (canakinumab), the systemic administration of the drug prevented the progression of DR and had promising results in reducing DR related macular edema; although, it had no effect on neovascularization [265]. Although these results are encouraging, there is still insufficient clinical evidence to recommend a fully anti-inflammatory treatment for DR patients; so far, the gold standard treatment is based on laser coagulation and intravitreal injections of anti-VEGF agents [266]. 

Glial cells are the main sources of cytokines in the CNS [33,34,35,37,267,268,269]. MGC, in addition to RPE, are the major sources of proinflammatory cytokines in DR [264]. MGC acquire a complex reactive phenotype in response to diabetes, with different expressions of 78 genes. Of these genes, 33% are associated with inflammation [270]. As reviewed by Coughlin and coworkers in 2017, under high glucose conditions, MGC are a source of damaging cytokines, such as IL-1β, interleukin 6 (IL-6), and TNF-α, which can be an important part of the chronic inflammation status of the diabetic retina [271]. However, what are the cellular mechanisms underlying the upregulation of cytokines in MGC that contribute to a shift to a pro-inflammatory phenotype in DR? Some new studies are focusing on this question to understand the cellular mechanisms that may help to improve the early treatment for DR. NF-κB, a well-known regulator of inflammation, is increased in MGC by hyperglycemia as an early response and has a great contribution to this pro-inflammatory response [167,170,171]. Tu and co-workers suggest that increase in ROS activates NF-κB, and Nrf-2 could block this pathway by controlling MGC oxidative stress [165]. In fact, the stimulation of Nrf2 dumps the augment of cytokines in MGC and diabetic retinas [168]. It has also been shown that advanced glycated elements, a common feature of diabetes, activate their receptors in MGC leading to glial dysfunction and increase in pro-inflammatory cytokines through MAPK signaling during diabetes [166,272]. In addition, hyperglycemia leads to Rho kinase (ROCK) activation, which increases Erk pathway signaling, resulting in an increase in pro-inflammatory production [273]. It was recently shown that C-myc, a protein considered a proto-oncogene, but also associated with inflammation [274], is expressed in MGC [275] and regulates pro-inflammatory cytokines release under hyperglycemic state (Figure 3). Interestingly, rat retinal Müller cells (rMC-1) cultures have increased C-myc expression after 4 h of high glucose exposure, while IL-1β, IL-6, and TNF-α levels augmented only after 24 h. Moreover, silencing C-myc in MGC prevented cytokine upregulation, demonstrating that this transcription factor is essential for initiation of the Müller cell inflammatory response in the diabetic context [276]. 

Likewise, once MGC adopt an inflammatory behavior, they can trigger inflammatory responses in other cells such as microglia/macrophages. CD40, an inflammation inducer, is increased in Müller, microglia, and endothelial cells in the retina of diabetic mice. In MGC, diabetes induced-CD40 activation is responsible for upregulating expression of inflammatory cytokines, such as TNF-α, IL-1β, NOS, and monocyte chemoattractant protein-1 (MCP-1 or chemokine ligand 2, CCL2) (Figure 3). Further, besides increasing these inflammatory mediators, CD40 activation also induces ATP release from MGC, leading to upregulation of TNF-α and IL-1β through P2X7 receptor activation in monocytic cells, which is a trigger for retinal inflammation [277]. Finally, ATP activation of P2X7 receptors forms large pores highly permeated by calcium, both in astrocytes [278] and in MGC [279]. Interestingly, the CD40-ATP-P2X7 pathway is also involved in both inflammation and death of retinal endothelial cells, promoting development of capillary degeneration and retinal ischemia [280]. Altered calcium signaling induced by overactivation of P2X7 receptors is a major step in the induction of neuronal and microvascular cell death under disease conditions such as ischemia-hypoxia and diabetes [281].

Together, these data suggest that MGC participate not only in inflammation amplification but also in triggering inflammatory responses in other cells. Interestingly, in transgenic diabetic mice that are knockout for P2X7 receptor, the upregulation of TNF-α, IL-1β, NOS, and ICAM was prevented [277]. Moreover, treatment with the nucleoside reverse transcriptase inhibitor lamivudine, a P2X7 receptor inhibitor, decreased progression of both neuronal and vascular pathology in DR, avoiding development of acellular capillaries and electrophysiological deficits, and the loss of GABA positive neurons [282]. In agreement with these results, Tu and collaborators demonstrated that melatonin prevented ganglion cell loss in diabetic mice through inhibition of Müller gliosis and prevention of increase in VEGF, TNF-α, IL-1β, and IL-6 in vitro and in vivo [165]. Furthermore, it was recently described that MGC treated with proliferative diabetic retinopathy vitreous humor obtained from diabetic patients had a higher expression of pro-inflammatory cytokines, demonstrating that the vitreous humor of patients with advanced DR had a potential for a proinflammatory response in MGC [208]. Altogether, these studies demonstrated that MGC are strongly associated with DR inflammatory insults and shed light on the manipulation of MGC inflammatory pathways as important therapeutic alternatives (Figure 3).

Altogether, data presented here show that while there is a great deal of evidence for the Müller role in retinal physiology, it is evident that the glial contribution in DR is still a field to be explored. A list of Müller roles in retinal physiology and alteration in diabetes condition is presented in Table 1. Further studies focusing on MGC participation in DR development are needed.

## 4. Future Directions

DR is a multifactorial progressive disease with accumulating vascular and neural damage that results in BRB breakage, microaneurysms, hemorrhages, edemas, neovascularization, and neurodegeneration. Currently, this disease is one of the major causes of visual impairment and blindness in the world [283,284]. Presently, available treatment consists mainly of intraocular anti-VEGF injections, which have side effects, such as cataracts and endophthalmitis. To further aggravate this situation, due to anti-VEGF short-life, frequent injections are necessary [285,286,287]. Other setbacks are that, although, this treatment slows down DR progression, it does not promote a cure and has limited use when vascular damage is advanced, and the disease is found in a late phase. Thus, it is evident that new alternative treatments are required to counteract DR progression at early stages, when the retinal tissue has not yet become severely compromised. 

Given the large number of functions performed by glial cells, it is expected that deficits in these cells have a major impact on nervous system functioning. Thus, modulation of glial function and reactivity state are important venues for the development of new preventive and therapeutic strategies for the care of neurodegenerative diseases [288]. 

As extensively discussed in this review, MGC play an essential role in retina neurovascular unit function and retinal homeostasis (Table 1; Figure 1B). During DR progression, some of MGC functions are severely impaired (Table 1), such as the ability to maintain BRB [289,290], control of glutamatergic [89,90,93] and ionic homeostasis [82,291], modulation of vascular function and neuronal integrity by VEGF [292], and maintenance of anti- and pro-inflammatory balance profile [108,192,271] (Figure 4).

Within the context of DR, different cocktails of molecules and compounds as potential drugs, both natural and synthetic, have been used to control MGC reactivity and improve retinal function. General antioxidant molecules have been tested as candidates for DR treatment in human trials, although with unsatisfactory outcomes. Molecules that could improve a specific cellular antioxidant response, such as Nrf2 signaling in MGC, for example, dh 404 or GSK inhibitors, might be therapeutic candidates for DR treatment [168]. Further, strong evidence shows that the Müller cell is an initial player in the pro-inflammatory retinal environment in DR [276,277]. Thus, control of the molecular pathways triggered by MGC could lead to new ways for preventing the early deficits found in DR, thus slowing down its progression. 

The next frontier of personalized medicine will be to understand the triggering mechanisms of how MGC might generate functional neurons to restore vision in mammals through reprogramming as a promising therapeutic strategy for human retinal diseases [293]. As for most neurodegenerative diseases, the main challenge of DR is to rescue the neuronal loss that usually occurs at a late stage of the disease. MGC may represent a potential candidate for this problem (Figure 5). The regenerative potential of MGC is well known in zebrafish, where retinal injury induces a proliferative response that could reconstitute all major retinal cell types and restore vision, a process named as reprogramming [294]. Previous studies have shown that MGC are closely related to retinal progenitors and act as latent neural stem cells that proliferate in response to injury and create a neurogenic environment, which possibly induces a response by novel retinal neurons. Uncovering this hidden potential could lead to new approaches to degenerative retinopathies, which are leading causes of irreversible visual impairment such as DR.

## 5. Conclusions

The main goal of this review was to highlight the need to consider MGC involvement in the progression of DR and potentially include these cells as targets for therapy development of DR. Either the identification of molecular pathways that control the function of MGC or the successful reprogramming of the fate of MGC should become a keystone for the development of new therapies addressing the progression of DR.

## Figures and Tables

**Figure 1 antioxidants-11-00617-f001:**
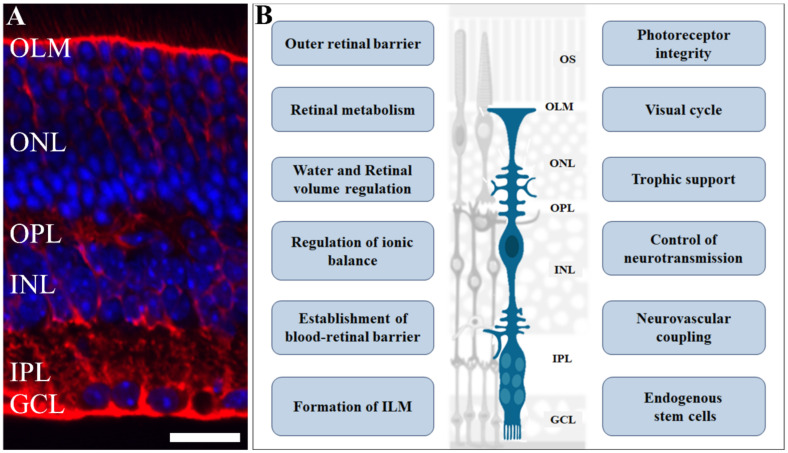
Role of Müller glia in retinal physiology. Müller glia cells have an elaborate radial morphology, characterized by vertical processes, that span throughout the retina and branches in outer and inner layers: (**A**) Photomicrograph of retinal section immunolabelled for glutamine synthetase (red), a Müller cell marker, counterstained with DAPI (blue), showing the retinal cells nuclei. (**B**) A scheme highlighting various functions and events regulated by Müller cells in the retina such as retinal volume and ionic balance; blood–retinal barrier formation and maintenance; neurotransmission control, trophic support for other cells, and are an endogenous source of stem cells. Scale bar= 20 µm. OS: outer segment; OLM: outer limiting membrane; ONL: outer nuclear layer; OPL: outer plexiform layer; INL: inner plexiform layer; IPL: inner plexiform layer; GCL: ganglion cell layer. Adapted from smart.servier.com (accessed on 1 April 2021).

**Figure 2 antioxidants-11-00617-f002:**
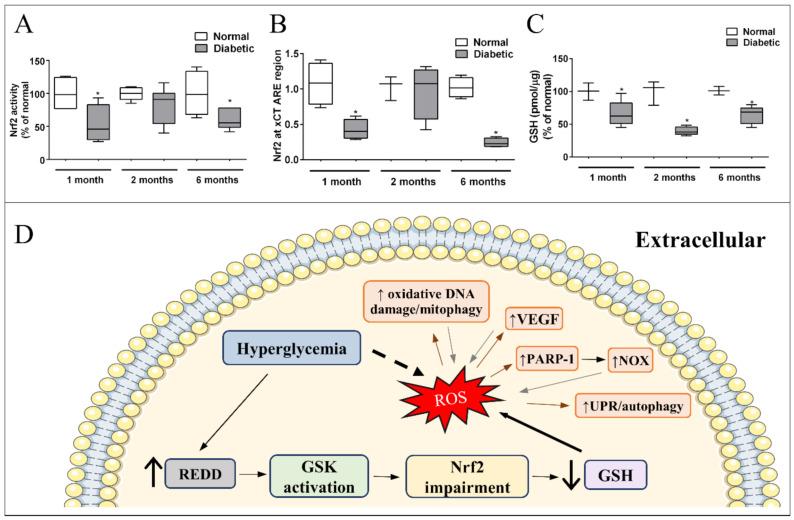
Signaling pathway underlying impaired Nrf2 function and effectors of excessive ROS in Müller cells in diabetes: (**A**) Nrf2 activity and (**B**) Binding to xCT ARE region, are decreased (↓) after 1 and 6 months of diabetes. (**C**) GSH is reduced in the retina of diabetic rats in early (1 month) and late periods (6 months). Modified from [237]. Copyright 2018 Elsevier. (**D**) Hyperglycemia induces an increase in REDD1 expression in Müller cells, which activates GSK. Once activated, GSK impairs Nrf2 function. As a result, there is a decrease in GSH levels and antioxidant defense capacity, resulting in higher oxidative stress in Müller glia. Direct effect of excessive ROS in Müller cells is indicated by brown arrows: increase (↑) in oxidative DNA damage/mitophagy, VEGF, poly [ADP-ribose] polymerase 1 (PARP-1), and unfolded protein response (UPR)/autophagy. All these responses intensify the production of ROS (gray arrows) enhancing oxidative stress in Müller cells. * *p* < 0.05 in comparison to normal.

**Figure 3 antioxidants-11-00617-f003:**
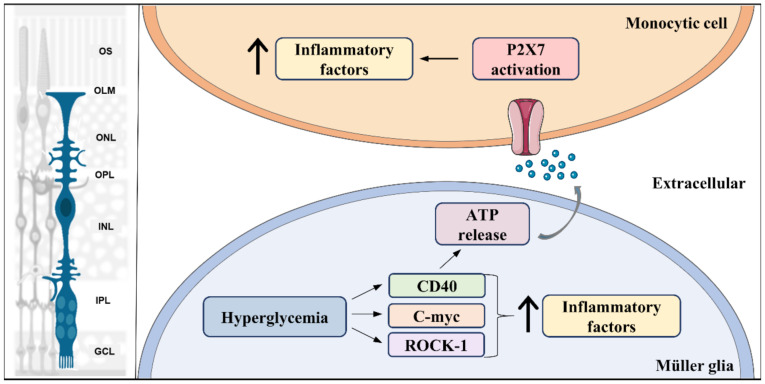
Pro-inflammatory signal pathways activated in Müller glia in diabetes. Hyperglycemia leads to activation of multiple pro-inflammatory signaling pathways, such as Rock-1, c-myc, and CD40, which results in an initial increase (↑) in production of pro-inflammatory factors in Müller glia. In addition, CD40 activation triggers ATP release by Müller glia, which acts as an inflammatory inducer in monocytic cells through its receptor, P2X7. Left part of the figure was adapted from smart.servier.com (accessed on 1 April 2021).

**Figure 4 antioxidants-11-00617-f004:**
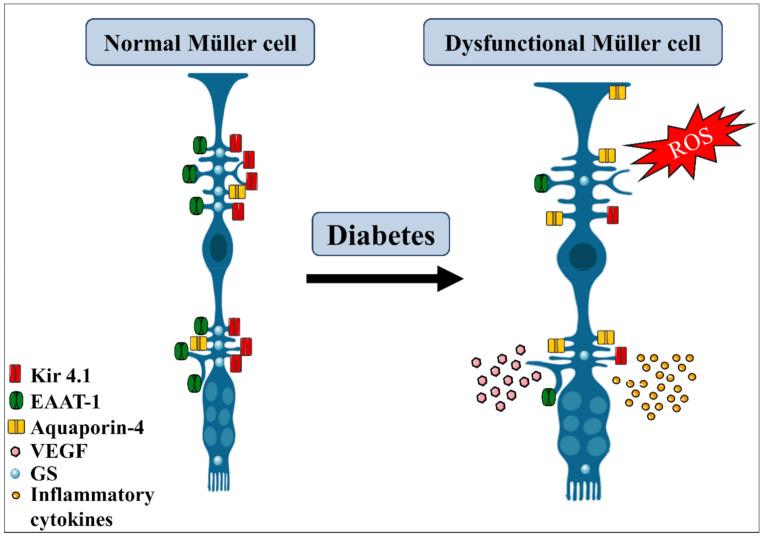
Diabetes effects on Müller glia. Diabetic condition triggers several morphological and metabolic alterations in Müller glia cells. They become hypertrophic, suffer increase in aquaporin-4, decrease the expression of Kir4.1 channel, glutamine synthetase (GS), and EAAT-1 transporters, thus affecting the regulation of the ionic balance and glutamate uptake. Under this condition, Müller glia also increase the production of reactive oxidative species (ROS) and the release of pro-inflammatory cytokines and VEGF, contributing to retinal damage. EAAT, excitatory amino acid transporter. Adapted from smart.servier.com (accessed on 1 April 2021).

**Figure 5 antioxidants-11-00617-f005:**
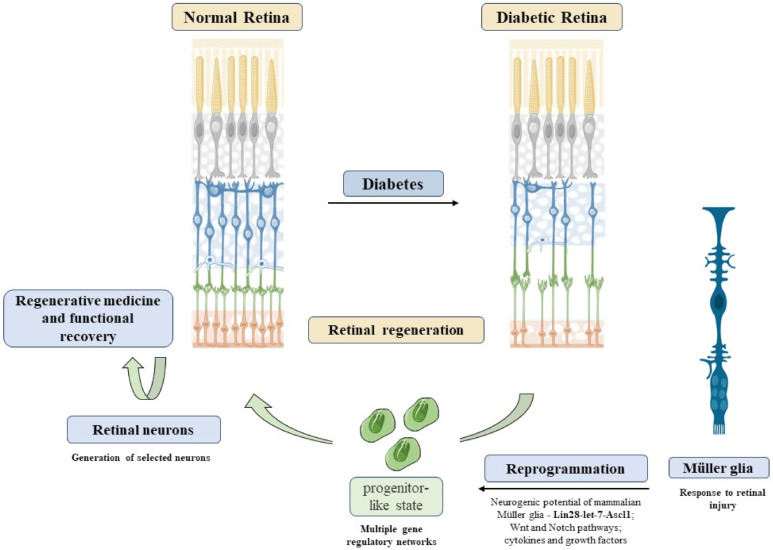
Reprogramming Müller cells in diabetic retinopathy. DR induces retinal neurodegeneration, an event difficult to recover. Alternatively, Müller reprogramming could act as a possible therapeutic option to restore visual capacity in DR patients. MGC reprogramming through activation of Lin28-let-7-Ascl1 genes, Wnt, and Notch pathways could lead to a progenitor-like state in these cells. Thus, these progenitors could be used to generate selected neurons, leading to retinal regeneration. Refs. [143,295,296,297] was adapted from smart.servier.com (accessed on 1 April 2021). Copyright 2016 Elsevier.

**Table 1 antioxidants-11-00617-t001:** Müller cell under retinal physiology and diabetic conditions.

**Physiological Role**	**Retinal Changes in Hyperglycemia/DM**	**Alteration of Müller in Hyperglycemia/DM**
**Regulation of water and retinal volume** [55,58,59,60,61,68]	**Increased thickness of INL and OPL** in patients [69] **Inhibition of TRPV4** associated with **decrease in the water diffusion** of diabetic mice retina [70] and **prevention of BRB breakdown** [71] **Alteration in AQP4 expression and/or localization** [50,72] **Increase in AQP1 and decrease in AQP6 and AQP11 expression** [73]	**Increase in cellular volume of Müller cells** [74,75] **Increase in AQP4 expression** [76,77,78] in Müller cell [75,79] **Swelling of Müller cells and cyst formation** [49,75]
**Regulation of ionic balance** [62,80]	**Reduction in Kir4.1** [72] **and potassium conductance** [74,81]	**Alteration in Kir4.1 and K_V_4 channels content and activity** [40,74,78,82,83,84]
**Control of glutamate neurotransmission** [40]	**Decrease in EAAT1 and glutamine synthetase expression** [85,86,87,88] **Reduced conversion of glutamate to glutamine and increase in retinal glutamate** [88,89,90] **Elevation in glutamate levels** in retinal cell culture, animal diabetic retina [87], and PDR patients vitreous [91,92]	**Decrease in EAAT1 function** [93] **and expression** [88] **Reduction in glutamate metabolism** [94]
**Control of GABA neurotransmission** [41]	**Increase in retinal GABA content** [95,96] **Enhancement of GABA levels** in PDR patients vitreous [91,92]	**Increase in GABA immunolabeling in Müller cells** [97]
**Survival of retinal neurons by providing neurotrophic factors and antioxidants** [41,98,99]	**ND**	**Neuroprotection** [100,101] **Increased protein content of neurotrophic factors, but also of inflammatory signals** [102]
**Blood–retinal barrier formation, maintenance and enhancement** [63,103]	**Increase in GFAP and reduction/redistribution in occludin expression in retinal vessels** [104] **Increase in adherent leukocytes in blood vessels** [105,106] **Increase in macromolecules leakage in outer BRB** [107]	**Decrease in occludin and ZO-1 expression** by **Müller cell-derived VEGF** [108] **Leakage resulting from Müller cell ablation** [109] **Protection against vascular alterations and angiogenesis** [110,111]
**Neurovascular coupling** [65,67]	**Alteration in neurovascular coupling** [66]	**ND**
**ILM formation** [41,54]	**ND**	**Alteration of ILM** [112] **Better prognostic after foveal sparing ILM peeli**ng in idiopathic macular hole [88]
**Neural regeneration** [113]	Intravitreal injected **adipose-derived stem cells differentiated into pericytes and integrate retinal vasculature** [114] Intravitreal injected **multipotent mesenchymal stromal cells prevented retinal ganglion cell loss** [115]	**ND**
Contribution to the **outer retinal barrier**, forming OLM [40,49,50]	**Alteration in occludin content and distribution** in diabetic retinas [49,116] **Alteration of OLM** in DME [117,118]	**ND**
Participation in the **visual cycle** [42,43]	**Alteration in visual cycle-related proteins** [119,120] **Involvement of visual cycle proteins with oxidative stress and vascular alterations in DR** [121]	**ND**
Contribution to **photoreceptors integrity** [51]	**Degeneration of photoreceptor cells** [122,123] **Alteration in morphology** [124,125,126,127] **and disorganization of inner and/or outer segments** [128] **Disruption of cone arrangement** [129]	**ND**
Support **high acuity vision in the foveola**, by expressing macular pigments [45,48,130]	**Decrease in macular pigment** [130]	**ND**
Impact on **retinal metabolism**: glycogen storage, primary use, and possible donation, of lactate, sparing oxygen to neurons [131,132,133,134]	**Alteration in glycogen and lactate contents** [135,136,137]	**ND**

ND: no description in the literature; AQP, aquaporin; DME: diabetic macular edema; DM, diabetes mellitus; INL, inner nuclear layer; OPL, outer plexiform layer; BRB, blood–retinal barrier; EAAT, excitatory amino acid transporter; PDR, proliferative diabetic retinopathy; ILM, inner limiting membrane; OLM, outer limiting membrane.

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
