# Peer review of "Contribution of Müller Cells in the Diabetic Retinopathy Development: Focus on Oxidative Stress and Inflammation"

_antioxidants, 2022, doi:10.3390/antiox11040617_

Round 1

Reviewer 1 Report

In the present manuscript, Carpi-Santos and colleagues summarized the contribution of Müller cells in the diabetic retinopathy development with a focus on oxidative stress and inflammation in a review submitted to the journal “Antioxidants”.

The manuscript was written most clearly with comprehensively collected data and well-organized views covering most current highlights on Müller functions in the development of diabetic retinopathy. It opens new insight into new target possibilities in this disease.  However, the manuscript needs to be improved and intensive discussions are required.

The link between neural damage, in particular vascular damage and oxidative stress/inflammation in Müller cells should be intensively discussed.

VEGF is required in physiological and pathological conditions in the retina. Differential discussion about the role of excessive VEGF expression in diabetic conditions might help reader to understand its crucial role in oxidative stress/inflammation correlated retinal vascular/neural damage.

As authors cited (in the Hammes 2018), oxidative stress is a main cause for cellular damage in diabetic retinopathy. Excessive ROS production induces DNA damage and activates PARP, which subsequently inhibits the activity of GAPDH, leading to the activation of four pathways including the accumulation of advanced glycation end products. Oxidative stress is surely also induced by different signal pathways in diabetic retinopathy. It would be extremely helpful if the authors can provide supportive data on whether oxidative stress/inflammation is a causal factor in the development of DR instead of an indicator.

In the NRF2 paragraphs, intensive discussion on vascular and neuronal damage correlated to oxidative/inflammation in NRF2 deficient/diabetic conditions will be needed.

Fig 2 and 3 with a zero SD/SEM in the controls are not suitable for statistical analysis. Thus, such figures should be deleted. In the figure 4, additional illustration of subsequent signaling, i.e. effectors of excessive ROS in Müller cells or other cells in the neurovascular unit might improve the manuscript.

Summarizing the data on interaction of Müller cells with other vascular cells and neural cells is required in the review. Additionally, authors may take care of how to describe the vascular cell loss in the diabetic retinopathy. Pericyte loss prior to EC loss, or vice versa?

In lines 573-605, the authors describe the effects of hypoxia and hyperglycemia. However, as authors stated, hypoxia is more or less strongly correlated with advanced DR that is out of the focus of the review.

In Fig. 7 the authors summarize the reprogramming Müller cells. However, a detailed and convincing discussion on the correlation between Müller cells as stem cells and oxidative stress/inflammation is missing in the manuscript.

Author Response

Please see our responses in the attached file.

Reviewer 2 Report

Carpi-Santos et al., have provided an extent review with data about the role of Müller cells in RD. I have enjoyed the text and the work is evident. Overall, the review is easy to read and follow and shows some interesting own results and a clear hypothesis. This review again highlights the potential of Müller cells to study RD. Mention some minor comments:

-Line 382. “4.1. Oxidative stress”. It might be interesting to mention the publication: “Oxidative and Endoplasmic Reticulum Stresses Mediate Apoptosis Induced by Modified LDL in Human Retinal Müller Cells” IOVS et al., 2012 by Wu et al. In this study, we hypothesized that extravasated and modified LDL, and the lipid oxidation products 7KC and 4HNE, cause Müller cell injury in diabetes and suggest that enhanced oxidative stress and ER stress are implicated.

-Figure 2. Mention in the manuscript the animal model used. Moreover, authors have measured the fluorescence intensity of DCFH-DA. Please, explain the meaning of the molecule and the implications.

-Line 550. Delete “retinal pigment epithelium” instead RPE.

-Line 561: Delete “Müller cells” instead MGC.

-Line 568. Explain the Müller cells cultures that mentioned. Are cell lines or primary? Human or mice cell line?

Author Response

(The authors gave the same response as above.)

Reviewer 3 Report

The Ms 1589663 by Carpi Santos et al. is a narrative review on the role of MGC in DR. Starting from an assumed supportive role for the retina, derangement of MGC function contributes to the development of DR and, probably, may offer targets for developing novel therapeutics. This main subject emerges clear from the abstract that sets well the scenario. However, the following 3 paragraphs distract the reader from the main subject and make the story very difficult to follow and mostly tedious. In fact, these latter paragraphs do not contribute at all to the main story of the article. With the scope to improve the impact of this Ms I would suggest the Authors to withdraw entirely the above numbered paragraphs to make the text more adherent to the title and abstract. The development of the article from there onward is then easy to read and can stand alone and this may probably stem from the original contribution of the Authors to the discussed subject. Probably, an additional improvement would originate from declaring the repository and temporal range in which the literature search has been carried out together with some other info (key words and search strings etc). This is not for changing the narrative nature of the review but to direct the reader with more methodological details.          

Author Response

(The authors gave the same response as above.)

Round 2

Reviewer 1 Report

The revision was adequate.